# Deep Eutectic Solvent-Mediated Electrocatalysts for Water Splitting

**DOI:** 10.3390/molecules27228098

**Published:** 2022-11-21

**Authors:** Chenyun Zhang, Yongqi Fu, Wei Gao, Te Bai, Tianyi Cao, Jianjiao Jin, Bingwei Xin

**Affiliations:** 1School of Intelligent Manufacturing, Wuxi Vocational College of Science and Technology, Wuxi 214028, China; 2College of Chemistry and Chemical Engineering, Dezhou University, Dezhou 253023, China

**Keywords:** deep eutectic solvent, metal catalyst, electrocatalysis, water splitting

## Abstract

As green, safe, and cheap solvents, deep eutectic solvents (DESs) provide tremendous opportunities to open up attractive perspectives for electrocatalysis. In this review, the achievement of DESs in the preparation of catalysts for electrolytic water splitting is described in detail according to their roles combined with our own work. DESs are generally employed as green media, templates, and electrolytes. A large number of hydrogen bonds in DESs result in supramolecular structures which have the ability to shape the morphologies of nanomaterials and then tune their performance. DESs can also serve as reactive reagents of metal electrocatalysts through directly participating in synthesis. Compared with conventional heteroatom sources, they have the advantages of high safety and designability. The “all-in-one” transformation strategy is expected to realize 100% atomic transformation of reactants. The aim of this review is to offer readers a deeper understanding on preparing DES-mediated electrocatalysts with higher performance for water splitting.

## 1. Introduction

Facing the serious challenges of energy crisis and environmental pollution, increasing attention is being focused on green energy, such as biodiesel, solar, wind, etc. [1,2]. H_2_ is becoming one of the most promising prospective green energy sources due to its high efficiency, zero emissions and renewability [3,4]. Electrocatalytic water splitting is a significant technological method of producing hydrogen. This reaction involves two half reactions: hydrogen evolution reaction (HER) and oxygen evolution reaction (OER) [5,6,7,8]. For this reaction, one of the most critical issues is exploring efficient electrocatalysts. In the process of preparing catalysts, obtaining high efficiency catalysts and simultaneously reducing environmental pollution is a core issue of research. Therefore, the application of green solvents and the realization of high atomic conversion of the reactants has always been the goal of people’s pursuit. In recent decades, ionic liquids (ILs) have been extensively studied [9,10,11]. ILs are composed of large organic cations and small organic or inorganic anions, thereby being liquid below 100 °C. Their special liquid structures and properties have made great achievements possible in catalytic chemistry. Nevertheless, the “greenness” of ILs is becoming challenged because their biodegradability, biocompatibility and sustainability are poor [12,13,14]. In addition, ILs are rather complex to purify, and expensive, resulting in certain restrictions in industrial production.

Under this background, deep eutectic solvents (DESs), a kind of alternative to ILs, have begun to draw people’s attention [15,16,17]. In 2003, Abbott and co-workers combined the choline chloride (ChCl, HOC_2_H_4_N^+^(CH_3_)_3_Cl^−^) (melting point, T_m_ ≈ 302 °C) and urea (T_m_ ≈ 133 °C) with a mole fraction of 1:2 and found that they finally presented as liquid with T_m_ ≈ 12 °C. They put forward the concept of “deep eutectic solvents” for the first time, opening up a new era in the application of eutectic solvents [18]. DESs refer to a two- or three-component combination of a hydrogen bond acceptor (HBA) and hydrogen bond donor (HBD) in a certain molar ratio [19,20,21,22,23]. The strong hydrogen bonds between HBA and HBD drive the mixture to liquid, and the freezing point of DESs is much lower than that of any component. HBAs are usually quaternary ammoniums (e.g., ChCl, tetra propyl ammonium bromide (TPAB), N_8881_Cl, etc.), phosphonium salts (e.g., P_14666_Cl, P_4444_Cl, etc.), hydrated metal salts (e.g., NiCl_2_∙6H_2_O, CoCl_2_∙6H_2_O, etc.), or Lewis acid metal salts (e.g., FeCl_3_, ZnCl_2_, etc.) [24,25,26,27,28]. Quaternary ammonium salts (R^1^R^2^R^3^R^4^N^+^·X^–^), especially ChCl, are the most common HBAs. HBDs are generally amides, carboxylic acids, polyols, and so on. DESs can be divided into five types [24], as shown in Table 1. Type I refers to the combination of quaternary ammonium salts and metal chlorides. Type II DESs are composed of quaternary ammonium salts and metal chloride hydrates, such as ChCl/NiCl_2_·6H_2_O and so on. Type IIIs consists of quaternary ammonium salts and HBDs (usually organic molecular components, such as amide, carboxylic acid, or polyol, etc.). ChCl/urea, ChCl/ethylene glycol (EG) as well as ChCl/glycerol are popular DESs. Type IVs are made of metal chloride hydrates and HBDs [25]. These four DESs occupy a dominant position, in which most of the studies have focused on Type III. At least one of HBA and HBD is an ionic species, which mainly interacts with strong hydrogen bonds, making the resulting mixture hydrophilic. However, a relatively new category of Type V DESs have been found, which consist of nonionic molecular HBAs and HBDs [26]. Although these compounds lack ionic contribution, a large number of hydrogen bonds make them retain the characteristics of DESs’ melting point. The clever design of HBAs and HBDs makes DESs “designer solvents”. The large number of hydrogen bonds between HBAs and HBDs induce DESs to possess special properties, just as for ILs. In addition, DESs behave with high ionic charge, dielectric constant and polarity, ideal conductivity and wide electrochemical potential window, nonvolatility, good thermal stability, etc. [27,28]. Therefore, DESs are considered as IL analogues [29]. Compared with ILs, DESs have advantages over ILs. They are low-cost, nontoxic, and biodegradable (Figure 1). In particular, they are easier to prepare. DESs have many preparation methods. The most common method includes heating and stirring the components of HBAs and HBDs until they form a uniform liquid. Other methods of preparing DESs involve vacuum evaporation, grinding and freeze drying. These preparation methods rarely require solvents, while sometimes requiring minimum water content. Therefore, there is no purification step, which is one of the biggest advantages over ILs. The economical and efficient preparation methods are more promising in industry. Since they have been discovered, DESs are recognized as cheap, renewable and green solvents. They have been rapidly applied in chemistry, biology, physics, engineering, and many other fields [30]. In the field of functional materials, DESs have attracted great attention. In 2004, Morris and his team carried out the preparation of inorganic materials in DESs and raised the concept of ‘‘ionothermal synthesis’’ [20]. This synthesis technology not only makes full use of the special modulation of DESs to inorganic micro/nanomaterials, but also avoids the danger of hydrothermal pressure because of the high stability and ignorable vapor pressure of DESs. Moreover, the favorable conductivity and wide potential window of DESs make them widely used as electrolytes to prepare inorganic nanomaterials by electrodeposition technology. DESs can be used as green media, templates, and sacrificial reagents in the preparation of micro/nanomaterials (Figure 1) [31,32]. Although the research on DESs is in infancy compared with ILs, the preparation of electrocatalysts assisted by DESs has become a new trend [33,34,35,36].

**Table 1 molecules-27-08098-t001:** Classification of DESs.

Type	Composition	Terms	Example	Ref.
Type I	R^1^R^2^R^3^R^4^N^+^·X^–^+MCl_x_	X is a Lewis base, generally a halide anion; M = Zn, Sn, Fe, Al, Ga, In, etc.	ChCl/ZnCl_2_	[24]
Type II	R^1^R^2^R^3^R^4^N^+^·X^–^+MCl_x_·yH_2_O	X is a Lewis base, generally a halide anion; M = Cr, Co, Cu, Ni, Fe, etc	ChCl/NiCl_2_·6H_2_O	[24]
Type III	R^1^R^2^R^3^R^4^N^+^·X^–^+HBD	HBDs include RCONH_2_, RCOOH, ROH, etc	ChCl/urea, ChCl/EG, ChCl/glycerol	[25]
Type IV	MCl_x_·yH_2_O+HBD	M = Zn, Sn, Fe, Al, etc; HBDs include RCONH_2_, RCOOH, ROH, etc		[25]
Type V	nonionic, molecular HBAs +HBDs		thymol/menthol thymol/lidocaine	[26]

**Figure 1 molecules-27-08098-f001:**
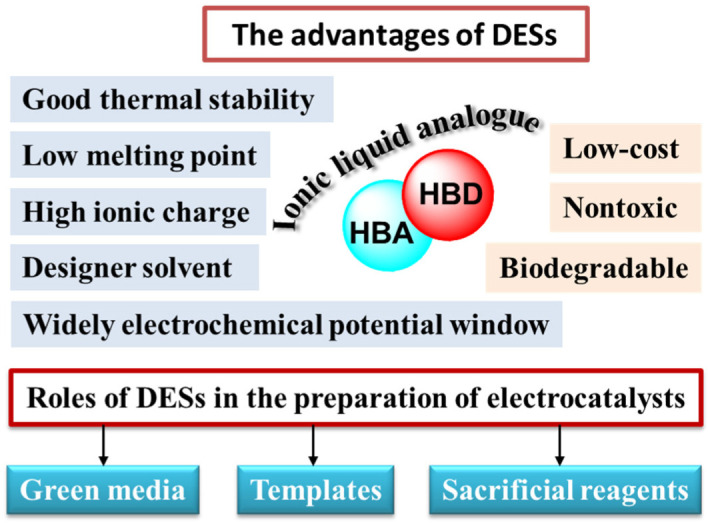
The advantages and roles of DESs in the preparation of electrocatalysts.

Compared with other solvents, DESs as media and sacrificial reagents can express the advantages in materials chemistry and catalytic chemistry in the following ways:(1)DESs are considered emerging green solvents and promisingly soft templates [37]. This is because DESs have special properties. Firstly, they exhibit fine solubility for metal salts, so that they are beneficial in synthesizing catalysts. Secondly, the low vapor pressure and high thermal stability of DESs allow the progress of reactions at high temperature and ambient pressure, avoiding the danger of high pressure. Thirdly, owing to the hydrogen bond, highly ionic strength and the viscosity of DESs, the microenvironment is different from that in conventional solvents. The supramolecular nature of DESs can tune structures and sizes of micro/nanomaterials.(2)DESs can serve as active reactants to prepare catalysts. Their designability makes them P, S, N, C or metals sources (such as Fe, Co, Ni and so on) to form phosphides, sulfides, nitrides, carbides or metal-based catalysts [38,39]. Compared with traditional heteroatom sources, they are safe and green. Moreover, owing to the influence of the special liquid structure of DESs during the reaction process, the obtained products show different structures and performance from those obtained from the conventional reactants. In addition, the conversion of DESs into electrocatalysts can reduce waste emissions and simplify operation processes.(3)The unique physicochemical characteristics of DESs result in different nucleation and growth mechanisms from those in conventional solvents through charge neutralization, changes in reduction potential as well as chemical activity, and determination of growth along the preferred crystallographic directions. In addition, DESs are able to change the activity order of metals, leading to some displacement reactions that cannot occur in aqueous solutions being undertaken in DESs [40].

To date, DESs have been widely explored in the preparation of catalysts for the electrolysis of water. Therefore, the progress in DES-mediated electrocatalysts for HER and OER is overviewed in detail in this review. This article is divided into four sections. Section 1 is the Introduction section, describing DESs and their advantages in the preparation of electrochemical catalysts. In Section 2, we describe DESs as both solvent and structure-directed reagents simultaneously for the synthesis of electrocatalysts. Section 3 discusses that DESs themselves are employed as reactive reagents to synthesize catalysts as sources of N, S, P, C, etc., while (hydrated) metal chloride-based DESs can provide a series of metallic elements, such as Fe, Co, Ni and Mn, etc. In Section 4, we draw conclusions from this review, while analyzing issues as well as prospects for further study on the evolution of DES-mediated electrocatalysts.

With the comprehensive development of DESs in various fields, a series of excellent reviews on DESs have been published focusing on food analysis, extraction, dissolution, organic synthesis, etc. [41,42,43]. Reviews with unique insights are also reported in the field of synthesis of functional materials [44,45]. However, up until now, no review has focused on the preparation of water splitting catalysts mediated by DESs. In this review, we made a detailed summary of this direction. We highlighted the advantages of DESs in the synthesis of inorganic electrocatalysts by comparing with traditional preparation methods. Meanwhile, our group is concerned with the 100% utilization of DESs and proposed an “all-in-one” DES strategy, which reduces emissions and minimizes pollution. We hope that readers will be able to follow the development trend in this field by reading this review, so that novel DESs may be designed and catalysts with novel structures and excellent properties may be prepared. Given the rapid development of this issue, the progress presented here covers about the period from January 2010 to October 2022.

## 2. Deep Eutectic Solvents as Both Green Solvents and Structure-Directed Reagents Simultaneously for the Preparation of HER and OER Electrocatalysts

With the rapid development of the concept of green environmental protection, DESs are one of the most investigated new green solvents [46,47]. Compared with traditional solvents, the low vapor pressure and high thermal stability of DESs allow the progress of reactions at high temperature and ambient pressure, avoiding the danger of high pressure. Although DESs are recognized as an alternative to ILs, DESs have advantages over ILs, such as low cost, natural origin, lower toxicity, and better environmental compatibility. Studies have shown that DESs are generally less cytotoxic than imidazolyl and pyridyl ILs, and can maintain high activity and unexpected stability of enzymes [48,49]. The safety of DESs makes them widely used in liquid-liquid, liquid-solid, and combined extractions [50]. In terms of the preparation of ILs and DESs, the synthesis of ILs usually needs a long time, which involves multiple synthetic steps with various reagents and organic volatile solvents. In addition, by-products and wastes are generated during the preparation of ILs. However, DESs can be simply prepared by a heating or grinding method with 100% yield. DESs are relatively simple and inexpensive to produce, and do not pose any major postpurification or disposal problems [51]. DESs also are promisingly soft templates. Owing to the hydrogen bond, highly ionic strength, and viscosity of DESs, the microenvironment is different from that in conventional solvents. The supramolecular nature of DESs can tune the structures and size of micro/nanomaterials. DESs are widely used in the preparation of electrocatalysts (Table 2).

Nickel-based materials have been promising electrocatalysts for HER and OER (Figure 2). Their preparation in DESs has been extensively researched in recent years. Studies have found that DESs as electrolytes are better than aqueous electrolytes due to their high ionic conductivity and wide electrochemical window [52,53,54]. Comparing aqueous electrolytes with DESs, many factors, including viscosity, conductivity and ionic property, are different. In DESs, NiCl_2_ · 6H_2_O displayed significant stable behavior due to coordination between DESs and Ni^2+^. In aqueous and DES, Ni disposition can be obtained with similar disposition rates affected by many factors. However, the Ni disposition obtained from ChCl/EG is considerably harder than that obtained from aqueous solutions, while presenting a different morphology. Ni nanoparticles obtained in DES via electrodeposition possessed smaller sizes and more uniform morphology than those obtained from an acetate buffer under similar conditions [55]. Therefore, Ni nanoparticles prepared by DES have high catalytic performance. Meanwhile, DES electrolytes are inexpensive, non-corrosive, compatible with electrode assemblies, and environmentally friendly, making them sustainable and cost-effective [56].

Further studies find that the compositions of DESs also affect the structure of the catalysts during the electrodeposition process. ChCl/urea and ChCl/EG DESs produced Ni films with different properties due to their different growth processes and assembling behaviors (Figure 2a). ChCl/EG has lower viscosity and higher conductivity than ChCl/urea. The diffusion coefficient of Ni^2+^ species in ChCl/urea is much slower than that in ChCl/EG. In addition, their coordination environment is also different. Therefore, the Ni films obtained from ChCl/EG possess different nanoparticles from those produced in ChCl/urea [57,58].

Different nickel salts behave differently in DESs, resulting in different nickel-based catalysts. For instance, replacing pure NiCl_2_ with the mixture of NiCl_2_ and Ni(NO_3_)_2_ led to obtaining Ni/Ni(OH)_2_ films on the substrate in ChCl/EG electrolyte via electrodeposition [59]. The reason should be that the reduction of NO_3_^–^ at a reducing potential generated OH^–^. Therefore, Ni(OH)_2_ was formed on the surface of Ni films. Clearly, the presence of NO_3_^–^ played a crucial role in the component of the product in the deposition process. The as-obtained Ni/Ni(OH)_2_ electrode required overpotentials of 110 mV for HER as well as 320 mV for OER to deliver a current density of 10 mA cm^−2^.

Although nickel itself is a promising Pt alternative metal for catalyzing HER, the strong adsorption capacity for hydrogen causes the desorption of H_2_ to be relatively slow. Doped S or P on Ni can tailor its electronic structure to optimize the intrinsic activity as well as the adsorption free energy of H_2_. Therefore, metal sulfides as electrocatalysts for H_2_O splitting have always been a research hotspot. Zhang et al. used an electrodeposition method to synthesize S-doped Ni microsphered films directly grown on Cu wire in ChCl/EG electrolyte [60]. They found that doping of S led to high amounts of oxygen vacancy on the surface. The influence of the ratio of Ni to S on the morphology and performance of the catalyst was investigated. NiS_0.25_ nanospheres possessed the biggest surface area among a series of catalysts with different Ni/S ratios. The synergistic effect of morphology and oxygen vacancies led NiS_0.13_ nanospheres to hold the highest electrocatalytic activity for HER. It showed a small overpotential of 54 mV to reach 10 mA cm^−2^ and continuous stability of 60 h in 1.0 M KOH solution.

More strikingly, DESs can alter the redox potential of metals, causing metal activity sequences different from those in aqueous solutions [61]. It is well known that the displacement reaction between Cu and Ni^2+^ is impossible without a chemical reducing agent in aqueous solution. However, experiments show that the redox potential of Ni^2+^/Ni in ChCl/EG is −0.154 V, while that of Cu^+^/Cu is −0.350 V at 353 K [61]. This phenomenon is attributed to the unique properties of DESs that provide different chemical environments for molecule solutions. Hence, the galvanic replacement reaction (GRR) between Cu and Ni^2+^ in ChCl/EG is thermodynamically feasible. This makes it possible to prepare Ni thin films on copper substrates by GRR. Using this theory, Zhang et al. prepared a series of nickel-based catalysts on the surface of copper foils (Figure 2b) [61]. Cu foils were put into ChCl/EG containing NiCl_2_. At the beginning of the reaction, Cu foils released electrons that could be oxidized to CuCl_3_^2–^, accompanied by the appearance of surface cracks. Rough and porous surfaces of Cu foils were formed and then acted as templates for the Ni films. With the increase in immersion time, Ni^2+^ was gradually reduced to Ni nanocrystallites and deposited on Cu foil. Finally, self-supported three-dimensional (3D) nanoporous Ni films were obtained. The obtained Ni films exhibited high HER catalytic activity with a low overpotential of 170 mV to arrive at 10 mA cm^−2^ and a relatively small Tafel slope of 98.5 mV dec^−1^ in alkaline media.

Based on the above work [61], this group then synthesized Ni_3_S_2_@Cu using GRR through adding thiourea as well as Cu foil into a ChCl/EG–NiCl_2_ system [62]. When the replacement reaction occurred between Cu and Ni^2+^, S was doped into Ni films to form S-doped Ni microsphere films on nanoporous Cu substrates (Figure 2c). It has been known that S can initiate structure changes with the superior exposed active position and electronic conductivity of Ni-based catalysts. Therefore, the addition of S could enhance the hydrogen evolution performance. The activated Ni_3_S_2_@Cu showed a high electrocatalytic HER activity with small Tafel slopes of 63.5 and 67.5 mV dec^−1^ in acidic and alkaline media, respectively. It required *η*_10_ (*η* represents the overpotential, *η*_10_ is the overpotential corresponding to a current density of 10 mA cm^−2^) of 91.6 mV in 0.5 M H_2_SO_4_ and 60.8 mV in 1.0 M KOH, respectively. Ni_3_S_2_@Cu exhibited higher catalytic activity than Ni nanocrystallites@Cu.

Similar to the preparation of nickel sulfides, nickel phosphides have been successfully prepared by adding appropriate P sources to DES-nickel salt systems using an ionothermal method [63], electrodeposition method [64,65], etc. The P-doped nickel films were formed on copper foil by electrodeposition in ChCl/EG containing a different Ni : P ratio (Figure 2d). The synergistic effect between P and Ni induced excellent HER catalytic performance. They could achieve *η*_10_ = 105 mV with a small Tafel slope (44.7 mV dec^−1^) when it had the suitable Ni:P ratio (1:0.056) [64].

**Figure 2 molecules-27-08098-f002:**
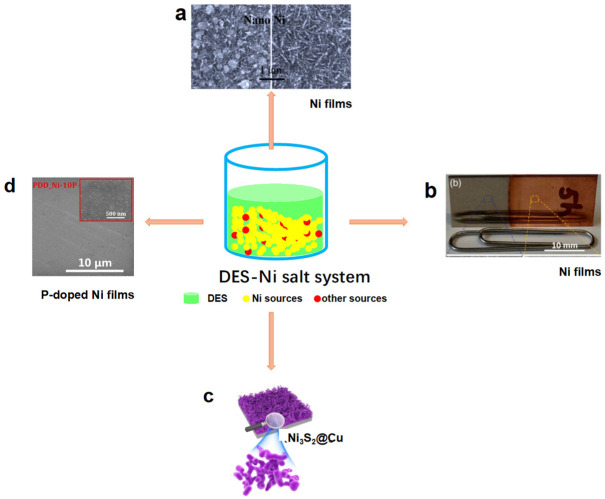
Electrochemical catalysts obtained from DES–Ni system: (**a**) Ni film. Reproduced with permission from ref. [57]. Copyright 2018 Elsevier. (**b**) Ni films. Reproduced with permission from ref. [61]. Copyright 2016 Elsevier. (**c**) Ni_3_S_2_@Cu. Reproduced with permission from ref. [62]. Copyright 2017 Elsevier. (**d**) P-doped Ni films. Reproduced with permission from ref. [64]. Copyright 2019 American Chemical Society.

The preparation of Ni-based bimetallic, even multimetallic, catalysts in DESs has drawn increasing attention. Adding the second metal salt, even several metal salts, into DES-nickel salt systems can produce Ni-based multimetallic nanomaterials, such as Ni-Mo/Cu [66], Ni–Mo–Cu [67], Ni–Cu [68], Ni–Co–Sn [69], Ni–Fe [70,71], Co_x_–Ni(OH)_2_/Cu foam [72], NiCo_2_O_4_ [73], (FeCoNiCuZn)(C_2_O_4_)·2H_2_O [74], and so on. The Ni-based multimetallic combination will regulate the position of the d-band and the electronic structure of Ni, resulting in the superior kinetics, selectivity, and stability. Therefore, they are high-performance catalysts for HER and OER. Taking Ni–Mo/Cu as an example, it required a low HER overpotential of 63 mV to deliver 20 mA cm^−2^ with a small Tafel slope of 49 mV dec^−1^ [66]. It also offered excellent OER electrocatalytic activity with an overpotential of 335 mV to reach 20 mA cm^−2^, as well as a moderated Tafel slope of 108 mV dec^−1^. Furthermore, this catalyst needed a cell voltage of 1.59 V to achieve a current density of 10 mA cm^−2^ for overall H_2_O splitting.

According to the preparation idea of metal sulfide, adding S sources into Ni-based bimetallic DES system will generate Ni-based bimetallic sulfides [75]. For example, adding thiourea into ChCl/EG containing NiCl_2_∙6H_2_O and FeCl_3_∙6H_2_O could synthesize 3D amorphous S–NiFe_2_O_4_/Ni_3_Fe films via the electrodeposition route [76]. It exhibited excellent OER catalytic activity with overpotentials of only 260 mV and 285 mV to deliver 100 mA cm^−2^ and 500 mA cm^−2^ in alkaline solution, respectively, which was equivalent to the reported Ni–S as well as Ni–Fe OER electrocatalysts at 100 mA cm^−2^. Similarly, ChCl/EG involving NiCl_2_, CoCl_2_ and thiourea system drove a 3D hierarchically porous Co,S–co–modified nickel microsphere array to grow on nickel foam (NiCo_x_S_y_/NF), still using electrodeposition technology [77]. The optimal NiCo_0.2_S_0.8_/NF required *η*_20_ = 65 mV for HER as well as *η*_20_ = 270 mV for OER in a 1.0 M KOH solution. Moreover, it could achieve overall H_2_O splitting at 1.57 V with high durability.

DES-assisted cobalt-based catalysts for overall water splitting are also receiving attention. They are synthesized by the similar synthetic protocol to DES-mediated Ni-based catalysts [78]. Generally, in DESs containing cobalt sources, the addition of other reactants, such as NaH_2_PO_2_·H_2_O [79], SeO_2_ [80], thiourea [81,82], NH_4_VO_3_ [83] and so on, can produce Co–P nanoparticles, Co–O/Co–Se hybrid films, Co–S films, CoV_2_O_6_ nanocrystals, etc. The ratio of Co and heteroatom in DESs directly affects the catalytic performance. For example, when P:Co was 1:1, Co–P@NF showed lower overpotentials of 62 mV and 320 mV at 10 mA cm^−2^ for HER as well as OER, respectively [79]. In addition, experiments have found that DESs can make difficult-to-obtain electrocatalysts more accessible. In ChCl/EG DES, 3D pompom-like Co–O and Co–Se hybrid films anchored on copper foils (Co–O@Co–Se/Cu) were prepared via potentiostatic electrodeposition process [80]. The incorporation of Se induced an interesting morphological change with the nanoparticles’ packed flat structure for the deposited Co/Cu (Figure 3a–c). The resulting high surface area and the porous architecture played key roles in significantly increasing the surface active sites and promoting the intrinsic catalytic activity. It has been known that cobalt-vanadium oxides are generally prepared through using vanadium oxides and cobalt salts by solid state reaction, hydrothermal or co-precipitation methods [84,85]. However, these approaches often require long synthesis time, high reaction temperature or result in amorphous products. Using DESs as the reaction media can overcome the above defects. The well-defined octahedral CoV_2_O_6_ nanomaterials were synthesized from 1:1 ChCl/malonic acid DES maline at a lower temperature [83], as shown in Figure 3d,e. This exhibited a high OER catalytic activity (*η*_10_ = 324 mV) with high durability for over 24 h (Figure 3f). Compared with the traditional solid state route, DES-assisted synthesis had two obvious advantages. Firstly, DES-assisted synthesis could be achieved at a lower reaction temperature. Researchers studied the generation mechanism of CoV_2_O_6_ in DES in detail. The IR spectrum showed that at 100 °C, CoCl_2_ · 6H_2_O and NH_4_VO_3_ first generated MCl_x_^–^ in DESs, in which metal ions were mainly adsorbed by DES clusters. The hydrogen bond framework of the DES served as both a homogeneous medium and a template for metal ion stabilization. The crystals were then obtained by calcination. A non-identifiable crystalline phase was observed at 400 °C. Increasing to 500 °C for 2 h obtained a clear crystalline phase of α–CoV_2_O_6_. Obviously, DES-derived strategy obtained products at a lower temperature than the solid state route (typically a reaction condition of 720 °C for 40 h using vanadium oxide and hydrogenated cobalt oxide). It was found that the formation of DES-assisted CoV_2_O_6_ began with the initial pyrolysis of maline initiated by metal salts. The reason was that the presence of Lewis acid of MCl_x_, especially CoCl_2_, could decrease the endo effect of the thermodestruction of DES, which initially triggered the thermal destruction of DES at an early stage. Light species obtained from the decomposition of DESs, such as CO, CO_2_, methane, ethane, etc., underwent further combustion with atmospheric oxygen, creating sufficient heat supply for oxide formation. The second advantage of a DES-assisted approach was that it could obtain well-defined octahedral morphology. This was because the supramolecular nature of the DESs, including hydrogen bonding and electrostatic interactions, reduced the overall energy required for phase formation. In addition, the added metal salt closely combined with the hydrogen bond network of maline to integrate the formed MCl_x_ into the hydrogen bond solvent structure.

Since Fe element has a strong affinity for oxygen, the catalytic performance of Fe-based electrocatalysts is much lower than that of Ni- and Co-based materials. Fe is usually designed to be one of the components of multimetallic compounds. It is found that even a trace amount of Fe can reduce the overpotential of transition metal catalysts [86,87]. In 2014, Gu et al. synthesized CoFe layered double hydroxide (CoFe–LDH) with large layer spacing in ChCl/urea using a “water injection” method [88]. With the rapid injection of water, urea could be decomposed into OH^–^, which could react with Co and Fe ions to obtain CoFe–LDH. In this process, the derivative species from DES were used as intercalators, contributing to the formation of large distances of interlayers. The prepared CoFe–LDH had good OER performance. Using a similar method, this group also synthesized Ni(OH)_2_ [89] and Co(OH)_2_ [90]. Recently, with a facile one-step electrodeposition process performed in ChCl/urea DES, Zhang et al. fabricated an Fe-doped bimetallic phosphate electrocatalyst, directly grown on a Cu substrate [91]. In this system, the iron content could be flexibly adjusted. The optimized Fe-doped bimetallic phosphate electrocatalysts exhibited efficient HER catalytic performance in acidic, alkaline, and neutral solutions. In addition, the catalyst also exhibited excellent OER performance in alkaline conditions. The authors considered that the superior performance of the catalyst was reflected in two aspects. Firstly, doping of Fe could tailor O active electronic properties in phosphate. Secondly, the combination of the unique 3D microsphere structure with a two-dimensional (2D) nanosheet internal architecture provided enough active sites to favor transfer kinetics.

After all, iron is the most plentiful transition metal element in the Earth’s crust, while having the advantages of good physicochemical properties and high safety. The study of single Fe-based catalysts has been concerned with the electrolysis of water. Recently, high-performance Fe-based catalysts have been successfully prepared in DESs. FeS_x_ films were fabricated via adding thiourea into ChCl/EG containing FeCl_3_ 6H_2_O. This required 340 mV to produce 10 mA cm^−2^ for OER [92]. Our group prepared DES-derived Fe_7_S_8_/Fe_2_O_3_ via the reaction between Fe powder and thioacetamide in ChCl/glycerol (1:2) DES [93]. The 2D porous nanosheets were attributed to the structure-directing function of ChCl/glycerol. Annealing at 400 °C under a nitrogen atmosphere generated a large amount of oxygen vacancies and improved the crystallinity of products. Under the synergistic effect of Fe_7_S_8_ and Fe_2_O_3_, it only required a low overpotential of 229 mV to produce a current density of 10 mA cm^−2^ for OER, while processed low Tafel slope of 49 mV dec^−1^ in 1.0 M KOH.

Metal Mn has increasingly attracted attention. Zhang’s group generated S–MnO_x_/Mn through electrodeposition and in situ electrochemical oxidation (Figure 3g–l) [94]. Compared with traditional aqueous solutions, DES has unique solvent characteristics, including low water activity, wide electrochemical window, highly ordered hydrogen bonds and large surface tension. These characteristics produced self-supporting Mn nano sheet arrays on carbon paper conductive substrates. Doping of S could change the phase composition and electronic structure of this catalyst, offering a highly porous architecture with plentiful exposed edges, while the electro-oxidation led to tailored phase composition. The introduction of S improved the catalytic performance. In addition, the perfect combination of MnO_x_ (high activity) and Mn (high conductivity) was conducive to reaction kinetics and electron transfer. As a result, S–MnO_x_/Mn offered a low overpotential (*η*_10_ = 435) with a Tafel slope of 89.97 mV dec^−1^.

**Figure 3 molecules-27-08098-f003:**
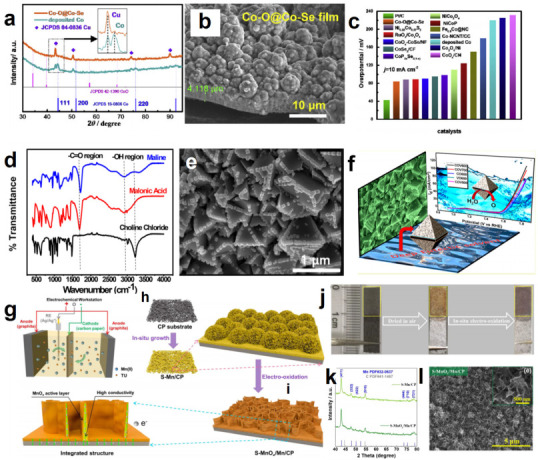
XRD patterns of Co−O/Co−Se (**a**), the cross section of the Co−O/Co−Se (**b**), comparison of the required potential at 10 mA cm^−2^ for the Co−O/Co−Se/Cu with other cobalt-based HER electrocatalysts (**c**). Reproduced with permission from ref. [80]. Copyright 2018 Elsevier; The IR spectrum of ChCl, malonic acid, and ChCl/malonic acid, respectively (**d**), SEM of CoV_2_O_6_ calcinated at 600 °C (**e**), diagram of octahedral CoV_2_O_6_ and its corresponding OER performance (**f**). Reproduced with permission from ref. [83]. Copyright 2018 American Chemical Society; Schematic of the three-electrode system for the preparation of active phases (**g**), direct growth of S−Mn nanoflake arrays in DES (**h**), in situ electrochemical oxidation to form S−MnO_x_/Mn (**i**), the optical image of preparation process of S−MnO_x_/Mn (**j**), XRD patterns (**k**) and SEM image (**l**) of S−MnO_x_/Mn. Reproduced with permission from ref. [94]. Copyright 2022 Elsevier.

Perovskites have attracted great interest as energy conversion materials due to their stability and structural tunability. Scientists have investigated a route to synthesize La-based perovskites using an environmentally friendly DES consisting of ChCl and malonic acid [95]. This method provided gram-scale and phase-pure crystalline materials and yielded high electrical active perovskites for the catalytic OER. Compared with perovskites prepared by other methods, the high activity of DES-derived LaCoO_3_ was attributed to the high concentration of the oxygen vacancy lattice, exhibiting current densities of 10, 50, and 100 mA cm^−2^ at corresponding overpotentials of approximately 390, 430, and 470 mV, respectively, and possessing a Tafel slope of 55.8 mV dec^−1^.

In addition to transition metal electrocatalysts, the DES-mediated noble metal (Pt, Rh, Ir, Ru et al.) electrocatalysts have also been studied [96]. They possess attractive properties, such as corrosion resistance, high conductivity and excellent electrocatalytic activity, making them ideal for applications. However, their application is limited by rare reserves and expensive value. Therefore, it is necessary to improve the efficiency of noble metals, such as by depositing precious metals on substrate. Pd was successfully attached to treated porous Ag surfaces (Pd@Ag), using ChCl/EG DESs containing PdCl_2_ as a solvent by GRR [97]. The DES could provide a unique solvent environment and reaction kinetics, which made a potential difference between Pd^2+^/Pd (0.951V) and Ag^+^/Ag (0.799V) and enabled Pd to grow on the porous Ag surface controllably. Then Pt was deposited on Pd@Ag by the electrodeposition method to generate Pt–Pd@Ag, which had wonderful HER performance in acidic, alkaline and neutral solutions.

Ru is an important platinum group metal. Ru films are usually prepared by electrodeposition. However, the electrolyte required for electrodeposition of metal Ru films is generally a toxic cyanide-based electrolyte. The overlapping of metal ion reduction and solvent reduction during electrolysis promotes the deposition of ethoxylated products on surfaces. In order to solve the above problems, Lee et al. successfully synthesized Ru nanoparticles with a rough surface using ChCl/urea DES as the electrolyte and stainless steel mesh as the substrate. DES could dissolve metals well without any additives [98]. The prepared Ru films had excellent onset potential (27.3 mV) and Tafel slope (97 mV/dec) for HER equivalent to standard Pt electrocatalyst [99].

It can be seen from the above achievements that the catalysts can be conveniently obtained by adding an appropriate source of heteroatoms via appropriate synthesis techniques. This simple design enables the realization of catalysts with multiple components. The synergistic effect between different components can tailor the structure of catalysts, improve their electronic structure, and thereby improving the catalytic activity.

In addition to the synthesis of metal nanomaterials in DESs, DESs can also be used to exfoliate layered catalysts. Layered compounds are an effective class of energy conversion catalysts due to a great deal of exposed edge sites on 2D nanomaterials [100,101]. However, these active sites tend to be less active due to aggregation by electrostatic and coordination interactions. An exfoliation technique is often used to increase and stabilize the edge active sites so that they enhance the electrocatalytic activity. DESs are found to be an ideal exfoliation agent because of the species in DESs, such as ammonium ions, which can be inserted into the layered crystals by some shear forces [102,103,104]. Assisted by DESs, single- or few-layer 2D materials can be obtained through mechanical grinding or chemical methods. The reactions between ammonium ions and layers result in forming a charged layered sheet that significantly weakens the Van der Waals force. The layered sheets no longer hold together and disperse as individual thin sheets in the solution. MoS_2_ is one of the typical layered compounds, which possesses a sandwich-like S–Mo–S layered structure stacked together via Van der Waals force. Mohammadpour studied the influence of DES composition on MoS_2_ stripping effect (Figure 4a–f) [105]. A series of sugar-based natural DESs and ChCl-based DESs were employed as intercalating agents. Natural DESs refer to ingredients that are abundant natural metabolites (including sugars, amino acids or organic acids). It was found that sugar-based natural DESs could strip MoS_2_, whereas most ChCl-based DESs were unsuccessful in inducing nanosheet production. Further comparing the stripping effect of different natural DESs, it was revealed that the natural DES of sucrose was better than that of either fructose or glucose. The reason was that sucrose-based DESs possessed larger molecular size than that from either fructose or glucose. Sucrose-based DESs formed large solvent clusters through hydrogen bonding. They could induce greater steric hindrance and were more prone to suppress the weak van der Waals interactions than those DESs involving smaller sized molecular components, when diffusing between the MoS_2_ layers and exfoliating them. As intercalating agents as well as environmentally friendly solvents, naturally sourced and nontoxic sugar-based natural DESs were firstly reported to achieve the exfoliation of MoS_2_ nanosheets with high yield. This study is likely to be extended to industry. The electrochemical testing results showed that the obtained 2H–1T MoS_2_ nanosheets exhibited an overpotential of 339 mV at the current density of 10 mA cm^−2^ and a long-term durability (2000 cycles) in acidic media for HER (Figure 4g,h). Subsequently, water was found to have the ability to modulate the molecular arrangement of DESs, inducing further regulating efficiency in MoS_2_ [106].

**Figure 4 molecules-27-08098-f004:**
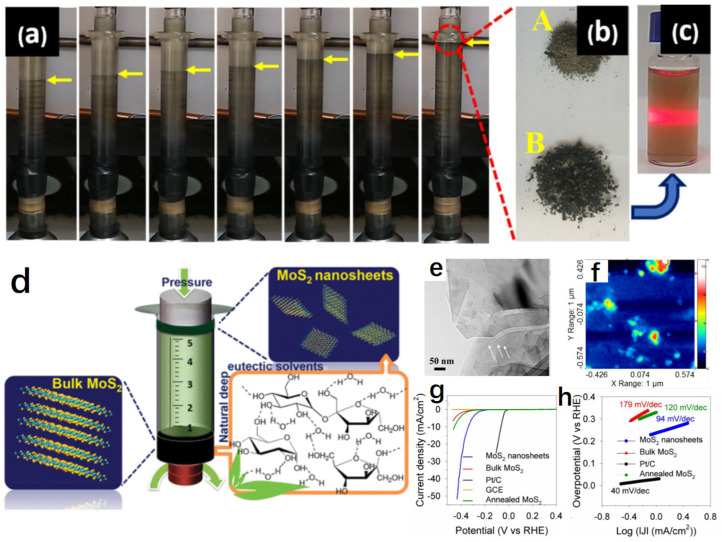
Longitudinal and gradual movement of the NADES/MoS_2_ mixture toward the syringe flange over time (**a**), photos of the raw exfoliation products (**b**), tyndall scattering of the raw exfoliation products (**c**), schematic diagram of MoS_2_ nanosheets synthesis process (**d**), TEM (**e**) and AFM (**f**) images of MoS_2_ nanosheets, polarization curves (**g**) and Tafel plots (**h**) of MoS_2_ nanosheets and other catalysts. Reproduced with permission from ref. [105]. Copyright 2018 American Chemical Society.

**Table 2 molecules-27-08098-t002:** Summary of HER, OER and Water Splitting Performance of Catalysts Involved in Section 2.

Catalyst	Applied DES	Preparation Method	Catalytic Performance								
			HER			OER			Water Splitting		
			Electrolyte	*η* (mV)@ Current Density (mA cm^−2^)	Tafel Slope (mV dec^−1^)	Electrolyte	*η* (mV)@ Current Density (mA cm^−2^)	Tafel Slope (mV dec^−1^)	Electrolyte	Potential (V)@ Current Density (mA cm^−2^)	Ref.
Ni	ChCl/Urea	Electrodeposition	1 M KOH	153@30	185	––	––	––	––	––	[57]
Ni/TiO_2_	ChCl/EG	Electrodeposition	1 M NaOH	––	122	––	––	––	––	––	[58]
Ni/Ni(OH)_2_	ChCl/EG	Electrodeposition	1 M KOH	110@10	83.9	1 M KOH	290@10	120.9	––	––	[59]
NiS*_x_*	ChCl/EG	Electrodeposition	1 M KOH	54@10	54	––	––	––	––	––	[60]
Ni	ChCl/EG	GRR	1 M KOH	170@10	98.5	––	––	––	––	––	[61]
Ni_3_S_2_	ChCl/EG	GRR	1 M KOH	60.8@10	67.5	––	––	––	––	––	[62]
Ni_3_S_2_	ChCl/EG	GRR	0.5 M H_2_SO_4_	63.5@10	91.6	––	––	––	––	––	[62]
NiP_x_	ChCl/EG	Electrodeposition	1 M KOH	105@10	44.7	––	––	––	––	––	[64]
Ni–P	ChCl/EG	Electrodeposition	1 M KOH	105@50	72.9	––	––	––	––	––	[65]
Ni–Mo	ChCl/EG	Electrodeposition	1 M KOH	63@20	49	1 M KOH	335@20	108	1 M KOH	1.59@10	[66]
Ni–Mo–Cu	ChCl/Urea	Electrodeposition	1 M KOH	93@10	105	––	––	––	––	––	[67]
Ni–Cu	ChCl/EG	Electrodeposition	1 M KOH	128@10	57.2	––	––	––	––	––	[68]
Ni–Co–Sn	ChCl/EG	Electrodeposition	1 M KOH	––	121	––	––	––	––	––	[69]
Ni–Fe	ChCl/EG	Electrodeposition	0.1 M KOH	316@10	62	––	––	––	––	––	[70]
Ni–Fe	ChCl/Urea	Electrodeposition	0.5 M NaOH	256@10	140.1	0.5 M NaOH	406@10	84.4	––	––	[71]
Co_x_–Ni(OH)_2_	ChCl/EG	Electrodeposition	1 M KOH	106@10	98.2	1 M KOH	330@100	126.7	––	––	[72]
NiCo_2_O_4_	ChCl/Glycerol	Calcining method	––	––	––	1 M KOH	320@10	67	––	––	[73]
(FeCoNiCuZn)(C_2_O_4_)· 2H_2_O	Polyethylene glycol (PEG)/Oxalic acid	Ionothermal method				1 M KOH	334@10	67.93			[74]
NiCo_x_S_y_	ChCl/EG	Electrodeposition	1 M KOH	65@10	62.5	1 M KOH	300@20	109	1M KOH	1.57@10	[75]
S–NiFe_2_O_4_/Ni_3_Fe	ChCl/EG	Electrodeposition	––	––	––	1 M KOH	260@10	35	1 M KOH	1.52@10	[76]
NiCo_x_S_y_	ChCl/EG	Electrodeposition	1 M KOH	65@20	54	1 M KOH	270@20	35	1 M KOH	1.57@10	[77]
Co	ChCl/Malonic acid	Electrodeposition	––	––	––	1 M KOH	350@10	76	––	––	[78]
P–Co	ChCl–EG	Electrodeposition	1 M KOH	65@10	69.2	1 M KOH	320@10	91.15	1 M KOH	1.59@10	[79]
Co–O/Co–Se	ChCl/Urea	Electrodeposition	1 M KOH	85@10	71.9	1 M KOH	340@10	67.6	1 M KOH	1.65@10	[80]
Co–S	ChCl/EG	Electrodeposition	1 M KOH	59@10	65	1 M KOH	307@50	66.4	1 M KOH	1.69@50	[81]
CoS_x_	Ethanedithiol/*n*–Butylamine	CO_2_–assited solution–processed method	––	––	––	1 M KOH	302@10	64.8	––	––	[82]
CoV_2_O_6_	ChCl/Malonic acid	Calcining method	––	––	––	1 M KOH	324@10	––	––	––	[83]
CoFe–LDH	ChCl/Urea	Water injection method	––	––		0.5 M KOH	Onset overpotential 510	––	––	––	[88]
Fe_x_Co_3–x_(PO_4_)_2_	ChCl/Urea	Electrodeposition	1 M KOH	108.1@100	30.3	1 M KOH	310@10	40.2	1 M KOH	1.62@10	[91]
Fe_x_Co_3–x_(PO_4_)_2_	ChCl/Urea	Electrodeposition	0.5 M H_2_SO_4_	128.8@100	42.4	––	_––_	––	––	––	[91]
Fe_x_Co_3–x_(PO_4_)_2_	ChCl/Urea	Electrodeposition	1 M Phosphate–buffered saline (PBS)	291.5@100	117.6	––	_––_	––	––	––	[91]
FeS_x_	ChCl/EG	Electrodeposition	––	––	––	1 M KOH	340@10	––	––	––	[92]
Fe_7_S_8_/Fe_2_O_3_	ChCl/glycerol	Calcining method	––	––	––	1 M KOH	229@10	49	––	––	[93]
S–MnO_x_/Mn	ChCl/EG	Electrodeposition and in–situ electrochemical oxidation	––	––	––	1 M KOH	435@10	89.97	––	––	[94]
LaCoO_3_	ChCl/Malonic acid	Calcining method	––	––	––	1 M NaOH	390@10	55.8			[95]
Pt–Pd@Ag	ChCl/EG	GRR and Electrodeposition	0.5 M H_2_SO_4_	28.1@10	31.2	––	––	––	––	––	[97]
Pt–Pd@Ag	ChCl/EG	GRR and Electrodeposition	1.0 M PBS	34.8@10	32.2	––	––	––	––	––	[97]
Pt–Pd@Ag	ChCl/EG	GRR and Electrodeposition	1 M KOH	23.8@10	32.5	––	––	––	––	––	[97]
Ru	ChCl/urea	Electrodeposition	0.5 M H_2_SO_4_	65.7@10	97	––	––	––	––	––	[99]
MoS_2_	A series of sugar–based natural DESs	Mechanical stirring	––	*––*	––	0.5 M H_2_SO_4_	339@10	94	––	––	[105]

## 3. Deep Eutectic Solvents as Reactive Reagents of Metal Electrocatalysts for HER and OER

The designability of DESs has prompted more enthusiasm to design them as active components which directly participate in forming catalysts [107]. This synthesis strategy has the following advantages. (1) DESs serve simultaneously as media, templates and reactants, making the reaction systems simple and reproducible. Meanwhile, the strategy which directly uses DESs as reagents is expected to realize 100% conversion of reactants. (2) HBDs and/or HBAs in DESs can be decomposed to release small molecules or ionic groups under appropriate stimulation, providing S, P, N, C, and even metal elements. Compared with traditional phosphorus and sulfur sources, and so on, DESs as heteroatom sources possess safety, green advantages, and other special properties. (3) The supramolecular structures of DESs have inheritance characteristics, which shape the structures and tune the performance of catalysts. Many kinds of DES-derived electrocatalysts, including metal-based catalysts, N-doped carbon materials, and metal and heteroatom co-doped carbon materials, have been synthesized and exhibited high performance (Table 2) by means of DESs as active species [108].

### 3.1. Deep Eutectic Solvents as Heteroatom Sources to Prepare Metal Electrocatalysts

The combination of metal and nitrogen-doped porous carbon is an effective strategy to improve the activity and stability of catalysts due to the improvement of conductivity [109,110]. Urea is an ideal nitrogen source for preparing N-containing catalysts [111]. Glucose, one of the most important and widely distributed compounds in nature, is a promising sustainable resource due to non-toxicity, renewability and low cost [112,113]. Glucose-based natural DESs are commonly used as precursors to prepare carbonaceous materials by heating DESs under N_2_. Pyrolysis of glucose/urea DES under a N_2_ atmosphere can produce 2D N,O-doped graphene [114]. Experimental results demonstrated that the DES calcining method was a comprehensive approach to produce novel carbon materials and adjust product morphology [115]. Furthermore, adding metal precursors into glucose/urea, DES could produce carbon flakes loaded with metal species via calcining them together without air, in which graphene flakes formed from DES acted as templates. When the products were further calcined under air, the graphene templates were removed. A series of metal oxides, such as La_0.5_Sr_0.5_Co_0.8_Fe_0.2_O_3_, Co_3_O_4_, NiCo_2_O_4_, RuO_2_ and Ba_0.5_Sr_0.5_Co_0.8_Fe_0.2_O_3_, were formed with low-dimensional hierarchical porous structure because graphene flakes formed from DES acted as template. These transition metal oxides could be applied for the OER. La_0.5_Sr_0.5_Co_0.8_Fe_0.2_O_3_ had the highest OER electrocatalytic activities (304 mV overpotential at a current density of 10 mA cm^−2^). Li further added ChCl to glucose/urea DES, obtaining ChCl/urea/gluconic acid ternary DES [116]. Cobalt nanoparticles supported on N-doped porous carbon (Co@NPC) were in situ synthesized through one-pot pyrolysis of this ternary DES and Co(NO_3_)_2_·6H_2_O. Co@NPC was employed as a bifunctional electrocatalyst for simultaneous HER and glucose oxidation reaction, a reaction instead of OER to promote H_2_ generation. Using urea or glucose components of DESs to provide N or C species is more effective than using pure urea or glucose. Firstly, DES ensures significant nitrogen contents due to the uniform incorporation of nitrogen into the solution phase of the synthesis process. Secondly, DES acts as a self-template to form an integral hierarchical structure composed of highly cross-linked clusters.

Our research group has engaged in in depth research to synthesize inorganic nanomaterials using DESs as reactive reagents. We rationally prepared DES-derived iron alkoxide based on the following considerations [117]. Firstly, it is found that the catalysts can slowly self-optimize under the condition of water electrolysis. Their structure and composition are transformed in situ [118]. Therefore, many reports have demonstrated the conversion of Fe-based catalysts during OER process. For example, tannic acid-nickel iron [119], FeSe vertical nanosheet array [120], etc., converted into FeOOH, a genuine active intermediate for electrocatalysis. Inspired by these achievements, we wonder whether iron alkoxide can catalyze water splitting and be converted to FeOOH under alkaline conditions, eliminating the steps to prepare FeOOH beforehand. Secondly, the preparation involving DESs is expected to shape special structures. The resulting product should be different from pure glycerol as a reactant as well as a medium. To explore the above problems, we chose ChCl/glycerol (1:2) as a benign reaction medium, in which glycerol was not only HBD but also the reactant [31]. After Fe(NO_3_)_3_ was put into ChCl/glycerol, a hydrogen bond was generated between NO_3_^–^ and C_3_H_8_O_3_, while Fe^3+^ was strongly chelated by C_3_H_8_O_3_. Thus, Fe(NO_3_)_3_ was easy to integrate into chcl/glycerol by hydrogen bonds as well as coordination bonds. Then the hierarchically 3D Fe-based organic-inorganic hybrid microsphere (Fe_DES) with large surface area was obtained at 150 °C for 11 h via solvothermal method (Figure 5a–d). Compared with pure glycerol, Fe_DES had superior structure and properties to the product from glycerol (Fe_Gly), which was amorphous and irregular. This stems from the special structure and properties of DES. The catalytic performance of Fe_DES was better than that of Fe_Gly. The overpotentials of Fe_DES only needed 280 mV and 300 mV to obtain 10 mA cm^−2^ and 20 mA cm^−2^, respectively, while the Tafel slope was small with good stability (Figure 5e–h). As expected, the oxygen evolution activity of Fe_DES improved significantly from the 1st to 40th indicated by the polarization curves. After 40 cycles, a stable polarization curve was reached. This activation process was attributed to its change in chemical composition and morphology in the alkaline electrolyte. It was found that the as-synthesized iron alkoxide was transformed into FeOOH during catalyzing of OER. The formation of FeOOH in situ afforded a sustainable catalytic performance. Meanwhile, the structure was reconstructed to a wrinkled surface which was more favorable than the original morphology, although basically maintaining the original microsphere and size (Figure 5i,j). The work has several innovations, including (1) developing a new method to prepare alkoxides, (2) firstly discovering iron alkoxide as a novel type of OER catalyst, (3) offering a guideline for the reasonable design of transformation-based catalysts in situ.

**Figure 5 molecules-27-08098-f005:**
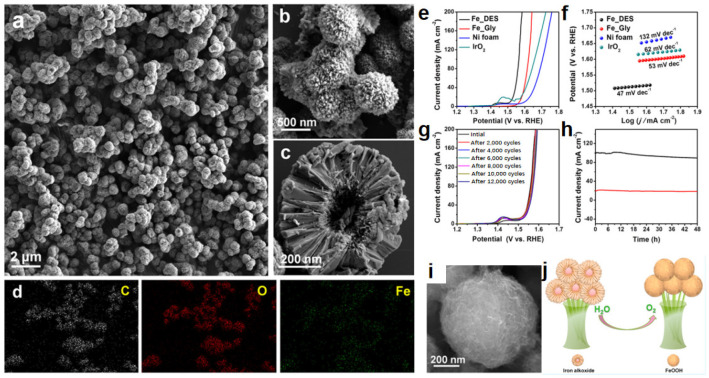
SEM images of low and high magnification (**a**–**c**) and EDX mapping (**d**) of iron alkoxide (Fe_DES), polarization curves (**e**), Tafel plots (**f**), polarization curves before and after cyclic voltammogram treatment (**g**), and chronoamperometry study (**h**) of Fe_DES, respectively. SEM image of Fe_DES after OER reaction (**i**), schematic illustration for the conversion from iron alkoxide to FeOOH during the OER process (**j**). Reproduced with permission from ref. [117]. Copyright 2019 American Chemical Society.

The participation of DESs can alter the conventional synthetic routes. The traditional preparation of metal sulfides usually requires a two-step method. Metal (hydro)oxides are preferentially prepared. Then they are vulcanized through adding sulfide sources [121,122]. This process is not only complicated but also poorly reproducible. The designability of DESs offers the possibility for simply synthesizing metal sulfides. Mu et al. devised a novel polyethylene glycol (PEG) 200/thiourea DES to prepare sulfides via a one-step solvothermal method [123,124]. Good solubility of DES in various metal salts led to a series of transition metal sulfides, such as CoS_2_, NiS_2_, Fe_3_S_4_ and Ni_2_CoS_4_, to be synthesized by adding metal salts to this PEGylated DES. As a multifunctional DES, the PEGylated DES acted as a solvent, shape-control agent and S source. Compared with traditional sulfuration approaches, this strategy was both economical and energy-saving by combining the solvothermal synthesis with the sulfuration process. The obtained NiCo_2_S_4_ exhibited a spherical sea urchin-like micro-/nano-structure composed of uniformly interconnected nanorods. It also showed a high surface area, abundant active sites, easy diffusion of electrolytes and oxygen gas, and strong structural integrity. Consequently, NiCo_2_S_4_ exhibited excellent OER properties. The overpotential was 337 mV to achieve 10 mA cm^−2^ with the Tafel slope of 64 mV dec*^−^*^1^ in 1.0 M KOH. Furthermore, NiCo_2_S_4_ presented long-term durability with little inactivation after 2000 cycles of successive operations.

From the achievements mentioned above, we know that DESs can participate in the reaction through their own decomposition. However, metals, as the central components of catalysts, would still need to be added in addition. To further simplify the synthesis process and achieve uniformity of element distribution, metal sources can also be designed as compositions of DESs. Hydrated metal chloride-based Type IV DESs have drawn great attention.

### 3.2. Hydrated Metal Chloride-Based Deep Eutectic Solvents as Metal Sources to Prepare Metal Electrocatalysts

Hydrated metal halide-based DESs are an important class of DES [125]. There are not only hydrogen bonds but also coordination between metal ions and oxygen donor ligands in DESs. They have special liquid structures. NiCl_2_·6H_2_O-based DESs have always been the focus of research. NiCl_2_·6H_2_O as HBA is often paired with carbon-rich HBD to form a DES, such as NiCl_2_·6H_2_O/PEG 200, NiCl_2_·6H_2_O/malonic acid and so on. They can provide both nickel and carbon species [126,127,128]. By sulfurizing or phosphating these kinds of DESs, carbon hybridized nickel sulfides or nickel phosphides are obtained. When NiCl_2_·6H_2_O/PEG 200 DES reacted with sublimed sulfur, a 2D NiS/graphene heterostructure was obtained via pyrolysis accompanied by sulfuration process in the tube furnace (Figure 6a,b) [126]. The ratio of graphene to NiS/graphene was tailored by simply changing the amount of PEG 200. In this kind of preparation strategy, both HBA and HBD participate in the reaction as reaction components, achieving the complete conversion of DESs. More importantly, the one-step procedure favored the interface coupling between graphene and NiS nanosheets, which significantly enhanced the electrical conductivity. The as-obtained NiS/graphene manifested excellent HER as well as OER activity in an alkaline solution. Furthermore, it showed a low cell voltage of 1.54 V at 10 mA cm^−2^, superior to the integrated IrO_2_ and Pt/C couples for water splitting. Similarly, the graphene oxide coated NiS_2_ (NiS_2_@GO) was obtained via pyrolysis of the mixture of NiCl_2_·6H_2_O/malonic acid DES and sublimed sulfur (Figure 6c–f). The unique DES supramolecular structure was conducive to the in situ growth of graphene in NiS_2_, and promoted the formation of gas channels. Optimized NiS_2_@GO catalysts required *η*_10_ = 57 mV and 294 mV toward HER and OER in an alkaline solution, respectively. Meanwhile, the battery voltage of 1.52 V could be obtained with being stable for 48 h. The excellent performance of water splitting may be attributed to the following aspects: (1) The interaction between different interfaces was conducive to the transmission of electrons; (2) The electrons of the catalysts could penetrate the shell films of GO to enhance the electrocatalytic reaction; (3) GO shell films could effectively protect NiS_2_ nanospheres from corrosion and prevent NiS_2_ from spreading out during electrolysis, inducing long-term stability and durability [127]. Ni_2_P@graphene was obtained through phosphating NiCl_2_·6H_2_O/malonic acid using NaH_2_PO_2_·2H_2_O [128]. It exhibited excellent activity with the potential of 1.51 V to arrive at 10 mA cm^−2^ for overall water splitting.

**Figure 6 molecules-27-08098-f006:**
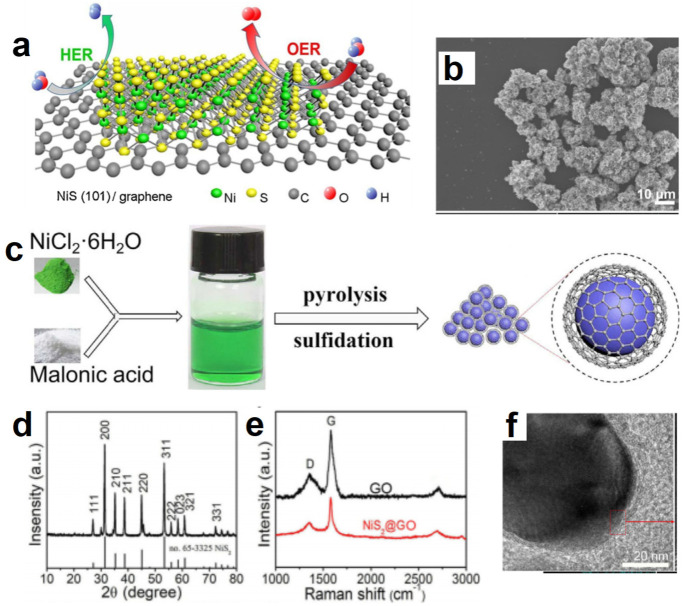
Schematic diagram of 2D NiS/graphene (**a**), SEM image of 2D NiS/graphene (**b**). Reproduced with permission from ref. [126]. Copyright 2019 Elsevier. Diagram of synthesis process and the microstructure of NiS_2_@GO (**c**); XRD pattern (**d**), Raman spectra (**e**) and TEM image (**f**) of NiS_2_@GO. Reproduced with permission from ref. [127]. Copyright 2020 Elsevier.

Metal halides and urea can also form a series of interesting DESs [129]. These kinds of DESs can introduce N, C, and metal elements into the catalysts. Based on this system, adding sulfur or phosphorus sources can produce N, C–co-doped catalysts by pyrolysis. For example, adding S source to NiCl_2_·6H_2_O/urea could produce 2D N–C/NiS_2_ nanosheets through sulfuration, which exhibits enhanced catalytic performance for water splitting [129].

Compared with NiCl_2_·6H_2_O, FeCl_3_·6H_2_O-based DESs as an iron source have less attention paid to them than synthesizing Fe-based micro/nanomaterials. To understand the behavior of Fe-based DESs, our group designed a system involving FeCl_3_·6H_2_O/urea DES and Ni foam (NF) with one-step synthesized flower-like NiFe LDH (Figure 7a–e) [130]. Dipping NF into FeCl_3_·6H_2_O/urea would precipitate a series of reactions. A redox reaction between Fe^3+^ and NF would make NF become oxidized to Ni^2+^. Then, Fe^3+^ and Ni^2+^ would construct NiFe–LDH on the surface of NF, with OH^–^ formed by the hydrolysis of urea. The reaction only took 30 s at 60 °C. Compared with the literature reports [131], reaction conditions in our dipping redox synthetic strategy had clear advantages, such as a mild environment, a simple operation, a low temperature, a rapid reaction, and so on. Moreover, it has been found that urea as HBD could form hydrogen bonds with Cl^−^ while coordinating with Fe^3+^ in FeCl_3_·6H_2_O/urea DES [125]. These interactions led to distinguishing structural characteristics. The design was more meaningful because urea was adsorbed on the surface of NiFe–LDH, achieving in situ modification of the catalyst resulting from the coordination effect of urea. The combination of urea could improve the flow of electrons and then facilitate the catalysis. Therefore, DES had four functions, medium, Fe^3+^ source, oxidant and in situ surface modifier, which realized “four-in-one” synthesis. To further explore the effect of DES on the characteristics of catalysts, we replaced FeCl_3_·6H_2_O/urea DES with FeCl_3_ aqueous solution as a control example. In addition to there being no urea modification of NF in an aqueous solution, the etching of NF in aqueous solution was more serious than that in a DES. Moreover, the mass loading of LDH on the NF was lower than that of a DES. In an alkaline medium, the nanoflowered DES-derived catalyst exhibited excellent electrocatalytic activity for OER with a potential of 1.39 V to achieve a current density of 10 mA cm^−2^, which was better than NiFe–LDH/NF obtained from a FeCl_3_ aqueous solution and most of the reported transition-metal catalysts. What is exciting is that the NiFe–LDH obtained from DES could achieve urea oxidation reaction, an anode reaction instead of OER, with the requirement of a potential of 1.32 V. NiFe–LDH/NF as cathode as well as anode could be implemented very well when considering water splitting and urea electrolysis. In this work, FeCl_3_·6H_2_O/urea DES was firstly employed to synthesize LDH nanomaterials. From the perspective of environmental protection and reactant utilization, this strategy is a green synthesis method because both HBA and HBD are active components when forming catalysts. FeCl_3_ as HBA is both an oxidant and a metal species. Urea as HBD can not only provide OH^−^ but also realize in situ surface engineering.

Inspired by the above achievements, researchers designed a new type of DES, NiCl_2_·6H_2_O/FeCl_3_·6H_2_O/urea/water [132]. This DES was heated by an ionothermal accompanied thermolysis reaction to obtain NiFe–LDH ultrathin nanosheets hybridized with N-doped carbon quantum dots. Each component in DESs contributed to the structure of a catalyst (Figure 7c–i). Owing to the existence of additional H_2_O in the precursor, urea pyrolysis would offer NH_4_^+^ and OH^−^. OH^−^ could react with a metal cation to form hydroxide. Meanwhile, the concentration of NH_3_ or NH_4_^+^ formed by urea pyrolysis would be high, which might insert and etch NiFe hydroxide, resulting in forming NiFe–LDH. The addition of water could affect the hydrogen-bonding network between HBAs and HBDs, thereby regulating the supramolecular structure and physicochemical properties of DES [133,134,135]. The in situ carbonization of the reaction systems led to the generation of N-doped carbon quantum dots, creating the disadvantage that carbon quantum dots need to be prepared in advance. A DES single precursor provided unique surface/interface energy for NiFe–LDH ultrathin nanosheets, making a mesoporous structure with rich defects. Meanwhile, DES ions were more easily inserted into the interlayer, so that the layer could be easily stripped to form ultra-thin nano sheets. Synergy of these factors was beneficial to the improvement of catalytic activity. This catalyst had excellent OER performance (*η*_10_ = 252, *η*_100_ = 311, *η*_500_ = 363 mV (*η*_100_ and *η*_500_ are the overpotentials corresponding to current densities of 100 and 500 mA cm^−2^, respectively)). This method provides a feasible and environmentally friendly strategy for the design and synthesis of other carbon hybrid ultra-thin LDH nanosheets with unique electrochemical properties [132].

This method can make both HBAs and HBDs participate in the reactions. To further realize the atomic 100% transformation of reactants, our group developed an “all-in-one” conversion strategy, which was consistent with the concept of green chemistry [136]. We directly heated CoCl_2_·6H_2_O/urea DES without adding any other reagents, resulting in the synthesis of 2D nanosheeted [Co(NH_3_)_4_CO_3_]Cl via the self-reaction of HBA and HBD, in which DES had been used simultaneously as a solvent, template, and reactant precursor (Figure 7j–n). We speculated that urea could be hydrolyzed to NH_3_ and CO_2_ under the action of a small amount of water existing in the hydrated metal salt. Then NH_3_ coordinated with Co^2+^ to form [Co (NH_3_)_4_]^2+^. The oxygen in the reactor could oxidize Co^2+^ to Co^3+^ under a high temperature and high pressure. Finally, [Co(NH_3_)_4_CO_3_]Cl was generated. Nanosheeted [Co(NH_3_)_4_CO_3_]Cl could transform into amorphous CoOOH during the activation process. It exhibited satisfactory catalytic efficiency (*η*_10_ = 291 mV, Tafel slope = 65 mV dec^−1^) as well as long-term durability (Figure 7o,p). The improved catalytic properties were attributed to rich Co^3+^ in [Co(NH_3_)_4_CO_3_]Cl. As a catalyst precursor, [Co(NH_3_)_4_CO_3_]Cl was converted into CoOOH during the activation process, an effective catalytic active component for oxygen evolution. The enriched Co^3+^ could not only effectively improve the adsorption energy of the H_2_O molecule, but also promote the deprotonation of the OOH* species to form O_2_. Meanwhile, the 2D activated [Co(NH_3_)_4_CO_3_]Cl increased the catalytic area with more exposed electrochemical active sites. The present all-in-one DES strategy opened the door for a reasonable design of DESs and advanced electrocatalysts from a sustainable perspective. By replacing urea in DESs with thiourea, CoCl_2_·6H_2_O/thiourea DES were obtained [137]. A defect-rich ultrathin N, S, and O tridoped carbon/Co_9_S_8_ was obtained by calcining the carbon cloth coated CoCl_2_/thiourea evenly. The catalyst had excellent electrocatalytic performance for HER in 1.0 M KOH, 1.0 M PBS, and 0.5 M H_2_SO_4_ electrolytes (*η*_10_ = 53, 103 and 68 mV, respectively).

**Figure 7 molecules-27-08098-f007:**
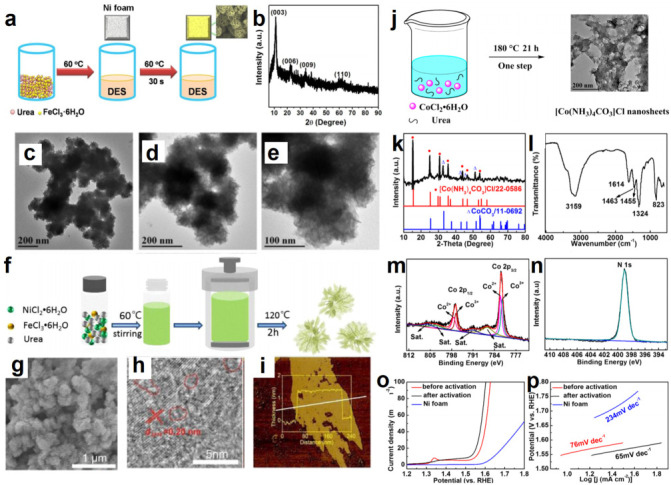
Schematic representation of the synthesis of NiFe–LDH/NF (**a**), XRD of NiFe–LDH (**b**), SEM images of NiFe–LDH (**c**–**e**). Reproduced with permission from ref. [130]. Copyright 2019 Wiley. Synthesis procedure and morphology of the NiFe–LDH/C nanosheets (**f**), SEM (**g**), HRTEM (**h**) and AFM (**i**) images of NiFe–LDH/C. Reproduced with permission from ref. [132]. Copyright 2021 Elsevier. Typical schematic illustration for synthesis of [Co(NH_3_)_4_CO_3_]Cl (**j**), XRD pattern (**k**), FT–IR spectra (**l**) of [Co(NH_3_)_4_CO_3_] Cl nanosheets, XPS survey spectrum of Co 2p (**m**) and N 1s (**n**) high resolution spectrum, polarization curves (**o**) and Tafel plots (**p**) of [Co(NH_3_)_4_CO_3_] Cl nanosheets and other catalysts. Reproduced with permission from ref. [136]. Copyright 2019 American Chemical Society.

Fe, Co, and Ni are called the “three musketeers” of electrocatalytic water splitting. To further improve the catalytic performance, it is desirable to combine these elements into one material [138,139,140]. However, it is difficult to achieve the combination of Fe, Co, and Ni components via conventional methods due to the different reducibility of Fe^3+^, Co^2+^ and Ni^2+^. However, using DESs can easily solve this limitation thanks to the excellent solubility of DESs. Recently, a novel four-component DES, hydrated trimetallic salt/L–cysteine, was designed [141]. HBA included three hydrated metal halides, FeCl_3_·6H_2_O, CoCl_2_·6H_2_O and NiCl_2_·6H_2_O. L–cysteine containing both –COOH and –NH_2_ functional groups acted as HBD. Directly calcinating this DES would produce advanced FeCoNi–based nitro-sulfide (FeCoNi–NS) by the self-reaction of the DES. The DES played multiple roles, including as complexing agent, shape-control agent, solvent, and source of metals N and S. The unique liquid structure of the DES gave the resulting product unique structural advantages, such as uniform dispersion of metal ions, high surface area and porous structure, etc. In addition, Fe, Co, and Ni components coordinated with each other to regulate the electronic structure. Therefore, FeCoNi–NS was conducive to efficient mass/charge transportation. The above characteristics were derived from FeCoNi–NS to improve the catalytic performance. The obtained FeCoNi–NS exhibited excellent OER performance, arriving at a current density of 10 mA cm^−2^ at an overpotential of 251 mV and low Tafel slope of 58 mV dec*^−^*^1^ in 1.0 M KOH.

Such a simple synthetic strategy is applicable to the manufacture of more multimetal hierarchical structures for energy storage and conservation. Mu et al. has made new progress in the field of DESs [142]. They designed a novel DES through mixing tetrabutylphosphonium chloride ([P_4444_]Cl), ethylene glycol, and five equimolar hydrated metal chlorides (FeCl_3_·6H_2_O, CoCl_2_·6H_2_O, NiCl_2_·6H_2_O, MnCl_2_·4H_2_O and CrCl_3_·6H_2_O). The hydrated metal chlorides and [P_4444_]Cl acted as HBA, while ethylene glycol served as HBD. This DES was used as a precursor to obtain metal phosphides with high entropy via a one-step in situ phosphorization under an inert atmosphere. The diffraction peaks of X-ray diffraction (XRD) examination as well as energy dispersive X-ray spectroscopy (EDX) elemental mapping revealed that a single crystal structure was produced, although it contained five metal species. This confirmed that the product was a high-entropy metal phosphide (HEMP). This phenomenon was attributed to the homogeneity and processability of DESs. As a bifunctional catalyst, this HEMP presented *η*_10_ = 136 mV for HER as well as *η*_10_ = 320 mV for OER in 1.0 M KOH. It was dramatically superior to the individual components and IrO_2_ catalyst (440 mV), suggesting its excellently catalytic performance.

As can be seen from the above results, the all-in-one DES strategy is an effective measure for preparing metal nanomaterials, and has the following advantages. (1) The supramolecules of DESs can adjust the morphology and catalytic performance of products. (2) The conversion of reactants is close to 100%, reducing emissions and minimizing pollution. (3) The pure DES reactants can lead to the uniform distribution of metal components and good reproducibility. (4) The one-pot process makes the operation simple. DESs as reactive reagents of metal electrocatalysts for HER and OER are summarized in Table 3.

## 4. Conclusions and Perspectives

DESs have made great developments possible in the area of electrocatalytic water splitting. As solvents and templates, DESs have good solubility, and can dissolve catalyst precursors well. DESs have the capability to regulate growth environment and mechanisms and can control the morphology and size of nanomaterials by their supramolecular structures. As reactants, DESs are safe and efficient when directly participating in the synthesis of catalysts. The designability of DESs allows their composition to be rationally designed to meet the different catalyst components. In the field of synthetic electrocatalysts, DESs have made great achievements. Using DESs can not only synthesize numerous catalysts (metal oxides, sulfides, phosphides, oxide perovskites, etc.), but also control the material structure, simplify the synthesis paths and improve the catalytic performance. The extensive prospects of DESs cause excellent research interest in the field of material synthesis. However, the studies of DES-based electrocatalysts still have some scientific problems to be solved.

### 4.1. Dialectically Understand the Greenness of DESs and Maximize Their Greenness through Reasonable Design and Effective Control Conditions

We should dialectically understand the advantages and disadvantages of DESs. At present, DESs are regarded as green solvents to some extent, and have been applied in many fields. However, there is growing evidence that DESs are not always green. Firstly, the hydrogen bond between HBAs and HBDs will be destroyed under certain conditions. We can recognize this point from the fact that DESs can be thermally decomposed to be sources of heteroatoms or metals, as shown in Section 3. In addition, the chemical, electrochemical and radiation decomposition factors can cause the instability of DESs, too. During the decomposition process of DESs, evaporation sometimes occurs, while toxic gases may be generated. This decomposition phenomenon will cause environmental pollution. Meanwhile, the loss of quality also goes against the atomic economy theory. In addition, DESs can react with other substances, which will cause corrosion to other substances. Furthermore, many DESs have been found to be toxic to biological organization in animals, plants, etc., to some extent [143,144,145,146]. So DESs are not completely green.

Therefore, it is necessary to explore strategies to improve the green degree of DESs. The designability of DESs, that is, the variable combination of HBDs and HBAs, can give novel DESs with different properties and compositions. Reasonable design composition and control of appropriate conditions will be effective methods to maximize their environmental friendliness [147]. This not only provides more opportunities for the preparation of novel catalysts but also tailors their stability and non-toxicity.

### 4.2. Requiring Further Understanding of the Structure-Composition-Performance Relationship

Until now, the role of DESs in the catalyst regulation mechanism and reaction process has been poorly understood. There exists still some randomness in the regulation of catalyst morphology and catalytic performance. It is necessary to further study the participation process of DESs in the nucleation and growth of catalysts, while further analyzing the compositional changes during the reaction process. A particular focus is the fundamental understanding of their structure-composition-performance relationships to realize the rational and controllable design of DES-based catalysts. Meanwhile, scientists should pay special attention to tracking the surface reconstruction process and identifying the real catalytic species under different test conditions, highlighting the significant differences in the corresponding electrochemical reconstruction mechanisms in order to further bridge the gap between laboratory scale research and large-scale application, promoting practical application and accelerating the progress of synergy between material science and engineering [148].

### 4.3. Exploring Research on the Preparation of Single-Atom Catalysts in DESs

Single-atom catalysis has now become one of the new frontiers and hotspots in the field of catalysis. Since single-atom catalysts have the “isolated active sites” characteristics of homogeneous catalysts and the “stable and easy to separate” characteristics of heterogeneous catalysts, they are considered to be expected to open up a new way of heterogenizing homogeneous catalysts and to become a link between homogeneous and heterogeneous catalysts. However, the preparation of single-atom catalysts using DESs has rarely been reported.

## Figures and Tables

**Table 3 molecules-27-08098-t003:** Summary of HER, OER and Water Splitting Performance of Catalysts Involved in Section 3.

Catalyst	Applied DES	Preparation Method	Catalytic Performance								
			HER			OER			Water Splitting		
			Electrolyte	*η* (mV)@ Current Density (mA cm^−2^)	Tafel Slope (mV dec^−1^)	Electrolyte	*η* (mV)@ Current Density (mA cm^−2^)	Tafe l Slope (mV dec^−1^)	Electrolyte	Potential (V)@ Current Density (mA cm^−2^)	Ref.
La_0.5_Sr_0.5_Co_0.8_Fe_0.2_O_3_	Glucose/Urea	Calcining method	––	––	––	1.0 M KOH	304@10	62.9	––	––	[115]
Co@NPC	ChCl/Urea/Gluconic acid ternary	Calcining method	1 M KOH	215@10	70	1 M KOH	430@10	87	1 M KOH	1.74@10	[116]
Iron alkoxide	ChCl/Glycerol	Ionothermal method	––	––	––	1 M KOH	280@10	47	––	––	[117]
NiCo_2_S_4_	PEG 200/Thiourea	Ionothermal method	––	––	––	1 M KOH	337@10	64	––	––	[123]
NiS/Graphene	NiCl_2_·6H_2_O/PEG 200	Calcining method	1 M KOH	70@10	50.1	1 M KOH	300@10	55.8	1 M KOH	1.54@10	[126]
NiS_2_/Graphene	NiCl_2_·6H_2_O/Malonic acid	Calcining method	1 M KOH	57@10	47	1 M KOH	294@10	54	1 M KOH	1.52@10	[127]
Ni_2_P/Graphene	NiCl_2_·6H_2_O/Malonic acid	Calcining method	1 M KOH	103@10	56.5	1 M KOH	275@20	56.2	1 M KOH	1.51@10	[128]
N–C/NiS_2_	NiCl_2_·6H_2_O/Urea	Calcining method	1 M KOH	78@10	63.4	1 M KOH	264@10	51.3	1 M KOH	1.53@10	[129]
NiFe–LDH	FeCl_3_·6H_2_O/Urea	Dipping–redox method	1 M KOH	160@10	42	1 M KOH	––	––	1 M KOH	1.61@10	[130]
NiFe–LDH/N–C	NiCl_2_·6H_2_O/FeCl_3_· 6H_2_O/Urea/Water	Ionothermal method	––	––	––	0.1 M KOH	363@500	49.8	––	––	[132]
[Co(NH_3_)_4_CO_3_]Cl	CoCl_2_·6H_2_O/Urea	Calcining method	––	––	––	1 M KOH	291@10	65	––	––	[136]
N,S,O–C/Co_9_S_8_	CoCl_2_·6H_2_O/Thiourea	Calcining method	1.0 M KOH	53@10	31	––	––	––	––	––	[137]
N,S,O–C/Co_9_S_8_	CoCl_2_·6H_2_O/Thiourea	Calcining method	1.0 M PBS	103@10	91.2	––	––	––	––	––	[137]
N,S,O–C/Co_9_S_8_	CoCl_2_·6H_2_O/Thiourea	Calcining method	0.5 M H_2_SO_4_	68@10	45.3	––	––	––	––	––	[137]
FeCoNi–NS	FeCl_3_·6H_2_O/CoCl_2_6H_2_O/NiCl_2_·6H_2_O/L–cysteine	Calcining method	––	––	––	1 M KOH	251@10	58	––	––	[141]
High–entropy metal phosphides	[P_4444_]Cl/Ethylene glycol/Five equimolar hydrated metal chlorides	Eutectic solvent method	1 M KOH	136@10	85.5	1 M KOH	320@10	60.8	1 M KOH	1.78@100	[142]

## Data Availability

Not applicable.

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
