# Peer review of "Deep Eutectic Solvent-Mediated Electrocatalysts for Water Splitting"

_molecules, 2022, doi:10.3390/molecules27228098_

Round 1

Reviewer 1 Report

In this review, the authors summarized the achievement of deep eutectic solvents in the preparation of catalysts for electrolytic water splitting. It is an interesting topic to review, I suggest a minor revision in order to better benefit the general audience of Molecules.

1. The more introduction of basic knowledge about DESs should be provided.

2. Is there any noble metal-based electrocatalysts involved in DESs?

3. Can the authors compare the methods with other emerging catalyst design strategies?

4. Four figures for a review article is not enough.

5. More discussion should be given when introducing the published works.

6. The following papers (Catalysts 2022, 12, 8, 928; Energy & Environmental Materials, 2022, e12441) are recommended to be cited for improving the manuscript.

Author Response

Nov. 7, 2022

Dr. Prof. Reviewer 1

Thank you very much for your letters on Thu., Nov. 3, 2022, which were about our manuscript entitled Deep Eutectic Solvent-Mediated Electrocatalysts for Water Splitting (molecules-2013785). We appreciate you very much because all of the comments are very valuable and helpful for us to make our manuscript better. We have revised the manuscript on the basis of the comments and suggestions. The changes are highlighted in yellow color in the revised manuscript. A list of changes and answers to the comments are given in the following pages. In the list, we marked the page and line number to facilitate reviewers to review.

On behalf of the coauthors, I hope that you are satisfied with the changes and look forward to a possible further publication process. In case of any question, please do not hesitate to contact me.

Thank you very much.

Sincerely yours,

Prof. Bingwei Xin

[email protected]

Dezhou University

Response to the comments and changes to the manuscript

Comments and Suggestions for Authors

In this review, the authors summarized the achievement of deep eutectic solvents in the preparation of catalysts for electrolytic water splitting. It is an interesting topic to review, I suggest a minor revision in order to better benefit the general audience of Molecules.

  1. The more introduction of basic knowledge about DESs should be provided.

Reply:

Thank you very much.

In Introduction Section, we described more basic knowledge about DESs, so as to make the article fuller and more popular. The added content are highlighted in yellow color in the reviewed manuscript. Please see page 1-4 scattered part highlighted in yellow color.

  1. Is there any noble metal-based electrocatalysts involved in DESs?

Reply:

Thank you very much.

 In the Section 2, we have added relevant literatures for noble metal-based electrocatalysts and described Pt-, Pd- and Ru-based electrocatalysts. These documents are marked as Ref. 96-99, respectively. Please see page 9-10 (the automatically generated line 381-401) part highlighted in yellow color.

  1. Can the authors compare the methods with other emerging catalyst design strategies?

Reply:

Thank you very much for your constructive comments. 

In Introduction Section, we have added the comparison about the methods with other emerging catalyst design strategies. Please see page 3-4 (the automatically generated line 99-121) part highlighted in yellow color.

  1. Four figures for a review article is not enough.

Reply:

Thank you very much for your constructive comments. 

In the revised manuscript, we have added 3 Figures, and now there are 7 Figures. In order to make the content richer, we also added Table 1 in the introduction section. These newly added or modified Figures or table are shown on: Table 1 (page 2-3), Figure 3 (page 9), Figure 4 (page 11), Figure 5 (page 14), Figure 6 (page 16) and Figure 7 (page 18).

  1. More discussion should be given when introducing the published works.

Reply:

Thank you very much for your constructive comments.

In the revised manuscript, we have provided more discussion for the published works. Please see the highlight part in yellow color in the whole revised manuscript.

  1. The following papers (Catalysts 2022, 12, 8, 928; Energy & Environmental Materials, 2022, e12441) are recommended to be cited for improving the manuscript.

Reply:

Thank you very much.

We have cited “Catalysts 2022, 12, 8, 928” as Ref. 36, while “Energy & Environmental Materials, 2022, e12441” as Ref. 148. Meanwhile, we have quoted the views of the literature “Energy & Environmental Materials, 2022, e12441”, which made the content of our article more profound. Please see page 20 last paragraph, the automatically generated line775-780).

Reviewer 2 Report

The manuscript ID: molecules-2013785 “Deep Eutectic Solvent-Mediated Electrocatalysts for Water Splitting”. In this review article, the author focused on the achievement of deep eutectic solvents (DESs) in the preparation of catalysts for electrolytic water splitting is described in detail according to their roles combined with their own work and highlights the advantages of DESs in the synthesis of inorganic electrocatalysts by comparing with traditional preparation methods.  Their presentation on review writing was not so exciting. I am recommending this review because only for few articles reported in this field. So, the topic is interesting for the referee. Therefore, I recommend publication only after minor revisions.

1) Page No:2; Line 45-46: “HBAs are usually quaternary ammoniums, phosphonium salts, or metal halides.” The author needs to give a reference here or explain clearly.

2) Page No:2; The author explained only the advantages of DESs and didn’t explain the disadvantages of DESs. Are there any drawbacks for DESs? If have, needs to include in this review article.

3) Page No.3; Line 90: the author has written “even metals”. Needs to write some examples.

4) Page No:11, Figures 3a, 3c, 3f, and 3g are not good. The author needs to include good figures.

Author Response

Nov. 7, 2022

Dr. Prof. Reviewer 2

Thank you very much for your letters on Thu., Nov. 3, 2022, which were about our manuscript entitled Deep Eutectic Solvent-Mediated Electrocatalysts for Water Splitting (molecules-2013785). We appreciate you very much because all of the comments are very valuable and helpful for us to make our manuscript better. We have revised the manuscript on the basis of the comments and suggestions. The changes are highlighted in yellow color in the revised manuscript. A list of changes and answers to the comments are given in the following pages. In the list, we marked the page and line number to facilitate reviewers to review.

On behalf of the coauthors, I hope that you are satisfied with the changes and look forward to a possible further publication process. In case of any question, please do not hesitate to contact me.

Thank you very much.

Sincerely yours,

Prof. Bingwei Xin

[email protected]

Dezhou University

Response to the comments and changes to the manuscript

Comments and Suggestions for Authors

The manuscript ID: molecules-2013785 “Deep Eutectic Solvent-Mediated Electrocatalysts for Water Splitting”. In this review article, the author focused on the achievement of deep eutectic solvents (DESs) in the preparation of catalysts for electrolytic water splitting is described in detail according to their roles combined with their own work and highlights the advantages of DESs in the synthesis of inorganic electrocatalysts by comparing with traditional preparation methods.  Their presentation on review writing was not so exciting. I am recommending this review because only for few articles reported in this field. So, the topic is interesting for the referee. Therefore, I recommend publication only after minor revisions.

1) Page No:2; Line 45-46: “HBAs are usually quaternary ammoniums, phosphonium salts, or metal halides.” The author needs to give a reference here or explain clearly.

Reply:

Thank you very much.

We have added the references and examples for “HBAs are usually quaternary ammoniums, phosphonium salts, or metal halides”. This sentence was changed to “HBAs are usually quaternary ammoniums (e.g., ChCl, tetra propyl ammonium bromide (TPAB), N8881Cl, etc.), phosphonium salts (e.g., P14666Cl, P4444Cl, etc.), hydrated metal salt (e.g., NiCl2∙6H2O, CoCl2∙6H2O, etc.), a Lewis acid metal salt (e.g., FeCl3, ZnCl2, etc.)[24-28].” Please see page 2 paragraph 1, the automatically generated line 52-53.

  • Page No:2; The author explained only the advantages of DESs and didn’t explain the disadvantages of DESs. Are there any drawbacks for DESs? If have, needs to include in this review article.

Reply:

Thank you very much.

We should dialectically understand the advantages and disadvantages of DESs. In the revised version, we added the disadvantages of DESs in Section 4 (Conclusion and Perspectives) in revised article. Meanwhile, we briefly describe that reasonable design composition and control of appropriate use conditions will be effective strategies to maximize their environmental friendliness. Please see page 20 paragraph 2, the automatically generated line 747-767.

  • Page No.3; Line 90: the author has written “even metals”. Needs to write some examples.

Reply:

Thank you very much.

We changed “even metals” to “while (hydrated) metal chloride-based DESs can provide a series of metallic element, such as Fe, Co, Ni and Mn, etc.” Please see page 4 paragraph 3, the automatically generated line 129-130. 

4)Page No:11, Figures 3a, 3c, 3f, and 3g are not good. The author needs to include good figures.

Reply:

Thank you very much.

In the revised manuscript, We have redone Figure 3 to make it better. In order to make the content richer, we also added to 7 Figures. These newly added or modified Figures are shown on: Figure 3 (page 9), Figure 4 (page 11), Figure 5 (page 14), Figure 6 (page 16) and Figure 7 (page 18).
